# Identification of Potential Drug Targets in *Helicobacter pylori* Using In Silico Subtractive Proteomics Approaches and Their Possible Inhibition through Drug Repurposing

**DOI:** 10.3390/pathogens9090747

**Published:** 2020-09-12

**Authors:** Kareem A. Ibrahim, Omneya M. Helmy, Mona T. Kashef, Tharwat R. Elkhamissy, Mohammed A. Ramadan

**Affiliations:** 1Department of Microbiology & Immunology, Faculty of Pharmacy, Egyptian Russian University, Cairo 11829, Egypt; dr_kareem@windowslive.com (K.A.I.); prof.tharwat.elkhamisy@gmail.com (T.R.E.); 2Department of Microbiology & Immunology, Faculty of Pharmacy, Cairo University, Cairo 11562, Egypt; mona.kashef@pharma.cu.edu.eg (M.T.K.); m_ramdan56@hotmail.com (M.A.R.)

**Keywords:** *Helicobacter pylori*, in silico, subtractive proteomics, drug targets, repurposing, chokepoint, database of essential genes, KEGG, BIOCYC

## Abstract

The class 1 carcinogen, *Helicobacter pylori*, is one of the World Health Organization’s high priority pathogens for antimicrobial development. We used three subtractive proteomics approaches using protein pools retrieved from: chokepoint reactions in the BIOCYC database, the Kyoto Encyclopedia of Genes and Genomes, and the database of essential genes (DEG), to find putative drug targets and their inhibition by drug repurposing. The subtractive channels included non-homology to human proteome, essentiality analysis, sub-cellular localization prediction, conservation, lack of similarity to gut flora, druggability, and broad-spectrum activity. The minimum inhibitory concentration (MIC) of three selected ligands was determined to confirm anti-helicobacter activity. Seventeen protein targets were retrieved. They are involved in motility, cell wall biosynthesis, processing of environmental and genetic information, and synthesis and metabolism of secondary metabolites, amino acids, vitamins, and cofactors. The DEG protein pool approach was superior, as it retrieved all drug targets identified by the other two approaches. Binding ligands (*n* = 42) were mostly small non-antibiotic compounds. Citric, dipicolinic, and pyrophosphoric acid inhibited *H. pylori* at an MIC of 1.5–2.5 mg/mL. In conclusion, we identified potential drug targets in *H. pylori*, and repurposed their binding ligands as possible anti-helicobacter agents, saving time and effort required for the development of new antimicrobial compounds.

## 1. Introduction

*Helicobacter pylori* is one of the most common infectious agents in the world, colonizing more than half of the global population, especially in developing countries where over 80% of their population are infected [1]. Infection with *H. pylori* is easily transmitted by the fecal-oral route causing chronic active gastritis, dyspepsia, gastric mucosa-associated lymphoid tissue lymphoma, iron deficiency anemia, idiopathic thrombocytopenic purpura, and is the main cause of peptic ulcer and gastric carcinoma [2]. *H. pylori* has been identified, since 1994, as a class-1 carcinogen [3]; it is the primary identified cause of gastric cancer, which is the second most common cause of cancer-related deaths worldwide [4].

Treatment of *H. pylori* infection involves the use of antimicrobial combinations, including clarithromycin, azithromycin, metronidazole, amoxicillin, tetracycline, and levofloxacin along with a proton pump inhibitor and bismuth salts, for a treatment period of 7–14 days, and even more [5]. Adherence to therapy, in *H. pylori* infections, is the biggest issue in treatment failures alongside resistance development to the antimicrobial agents [6]. This is further complicated by the risk of re-infection, especially in areas with high *H. pylori* infection prevalence [7].

Management of *H. pylori* infections has posed an economic burden on healthcare systems all over the world. It is becoming more challenging due to the emergence of antimicrobial resistance. The rate of resistance to levofloxacin, metronidazole, and clarithromycin have reached 15% in over 66,000 *H. pylori* isolates, according to a meta-analysis systematic study conducted in 2018 [8]. Clarithromycin resistant *H. pylori* is classified, by the World Health Organization, as a high priority pathogen for the development of new drugs [9]. 

The wide spread bacterial resistance, with the burden it poses on healthcare providers, highlights the concept of drug repurposing as a strategy to identify new uses for pre-existing drugs [10]. Anthelmintic, anti-inflammatory, anti-cancer, anti-psychotic drugs, and statins have been confirmed to possess anti-bacterial action; this is an approach that can save time, cost, and effort, as the safety and the pharmacokinetics properties of these agents are already documented [11]. In addition, novel approaches for antimicrobial drug discovery are now ongoing, based on genomics, proteomics, metabolic pathway analysis, essential gene analysis, and reverse docking [12,13]. They depend mainly on mining the genome/proteome sequence of the pathogen using many of the available bioinformatic tools, through different subtractive channels, such as similarity to human proteome, essentiality to the pathogen, and sub-cellular localization prediction to identify potential targets [14]. 

In silico subtractive approaches, besides being time saving, are cost effective in drug research and development stages [15]. These approaches were used to identify potential drug targets in many pathogens, including *Acinetobacter baumannii*, *Burkholderia pseudomallei*, *Mycobacterium avium*, *Neisseria meningitides*, *Pseudomonas aeruginosa*, *Staphylococcus saprophyticus*, and *Streptococcus pneumoniae* [16,17,18,19,20,21,22]. Similarly, several potential drug targets in *H. pylori* were also identified [12,23,24,25,26]; however, none of these studies predicted possible drug binding ligands, except for a study that proceeded to druggability analysis [27].

In our study, subtractive proteomics approaches were used to identify putative drug targets in *H. pylori*, through mining different available databases using strict inclusion and exclusion criteria, to yield potential host-safe drug targets. These targets are non-homologous to human, essential for the survival of *H. pylori*, cytoplasmic, less prevalent in common organisms of gut flora, and can act as broad-spectrum drug targets. Drugbank was searched for the possible repurposing of available non-antibiotic drugs as binding ligands. We were able to identify 17 protein targets with 42 binding ligands available at the drugbank. 

## 2. Results

Three approaches were used to search the proteome of *H. pylori* for druggable targets. The analyzed protein pools were chokepoint proteins, metabolic pathway proteins or essential proteins. They were further subjected to different analysis steps, in each approach, to determine the druggable targets (Figure 1). 

### 2.1. Chokepoint Proteins Analysis

A Chokepoint reaction is either a unique consumer or producer of a metabolite and is regarded as a good drug target [28]. Searching the chokepoint reactions of the *H. pylori* ATCC 43504 strain in the BIOCYC database, after the exclusion of proteins found in humans and reactions that are catalyzed by more than one enzyme, resulted in 483 chokepoint reactions: 222 reactions on the consuming side and 261 reactions on the producing side. Upon analyzing these reactions, spontaneous and repeated reactions, along with those which had no identifiable enzymes in the database, were excluded, leaving 103 and 120 reactions in the consuming and producing sides, respectively. Eighty reactions were chokepoints on both sides, and the net yield was 143 proteins.

Proteins having significant similarity with human proteome (*n* = 50), identified by the National Center for Biotechnology Information (NCBI) BLASTp tool, were excluded. This resulted in 93 non-homologous targets. When these targets were searched for in the database of essential genes (DEG), only 21 proteins with percentage identity >95 were found to be essential for *H. pylori*. Prediction of the localization of these essential proteins, using the PSORTb cellular prediction tool, resulted in the exclusion of three outer membrane-proteins. All the remaining cytoplasmic proteins (*n* = 18) were checked for conservation in *H. pylori* strains, available at the NCBI database (Appendix A), where all the tested proteins were conserved with significant similarity of >200 alignment scores. When these proteins were checked for homology to the proteome of common gut flora organisms, retrieved from a similar in silico study [29], eight proteins had significant similarity with an alignment score >200 and were, therefore, excluded from the analysis (Appendix A).

Non-antibiotic binding ligands for the remaining 10 proteins were searched for at the drugbank [30]; only six proteins had potential binding ligands, and they included: 3-dehydroquinate dehydratase (AroQ), carboxy-S-adenosyl-L-methionine synthase (CmoA), 4-hydroxy-tetrahydrodipicolinate reductase (DapB), D-alanine-D-alanine ligase (Ddl), geranyl diphosphate synthase (IspA), and riboflavin synthase (RibC), as listed in Table 1.

### 2.2. Metabolic Pathway Analysis 

In this approach, the initial protein targets were those involved in the metabolic pathways of *H. pylori*. These pathways were retrieved from the Kyoto Encyclopedia of Genes and Genomes (KEGG) [31,32,33] database, where 93 pathways for *H. pylori* strain 26695 were available, comprising 1672 proteins. After the removal of repetitions and non-identifiable enzymes, only 606 proteins remained; 402 proteins showed no significant similarity when compared to human proteome at NCBI BLASTp (Appendix A).

The human non-homologous proteins (*n* = 402) were checked in DEG for essentiality, where 86 proteins were essential for *H. pylori*. Sub-cellular localization prediction yielded 12 proteins with unknown localization, one periplasmic-, 13 cytoplasmic-membrane-, and 60 cytoplasmic-proteins. Among the cytoplasmic proteins, eight showed alignment scores below 200 with other *H. pylori* strains available at the NCBI database, and were, therefore, excluded. Upon comparing the remaining 52 proteins with common gut flora: 17 proteins were excluded because they had >200 alignment scores; seven proteins showed no significant similarity to any of the compared gut flora organisms, and one protein showed only 40–50 score; 27 proteins had similarity scores between 50 and 200 (Appendix A). 

Only 16 protein targets, out of the selected 35 proteins, were druggable (Appendix A; Figure 1). RplV was excluded, as lefamulin and quinupristin antibiotics were its binding ligands. These targets included all proteins retrieved from chokepoint analysis (except CmoA), besides shikimate dehydrogenase (AroE), chemotaxis protein Y (CheY), DNA polymerase III subunit β (DnaN), ribose-5-phosphate isomerase B (LacA/Rpi), acyl- [acyl-carrier-protein]—UDP-N-acetylglucosamine O-acyltransferase (LpxA), aspartate 1-decarboxylase proenzyme (PanD), 50S ribosomal subunit L5 (RplE), 30S ribosomal subunit S6 (RpsF), 30S ribosomal subunit S10 (RpsJ), and type IV secretion system ATPase (TrbB/VirB11_2). They all had >200 alignment scores to common pathogens, except for PanD, LacA/Rpi, RpsF, and RpsJ, which had an alignment score <200 (Table 1). The general and related metabolic pathways of these targets are summarized in Figure 2.

### 2.3. Essential Proteins Analysis 

Three hundred twenty-three essential proteins for *H. pylori* strain 26695 were retrieved from DEG. This was followed by searching the drugbank for possible binding ligands, where 100 proteins were druggable. After excluding RplV (having an antibiotic ligand) and 61 proteins that showed significant similarity to human proteome, 38 proteins were retrieved. Their sub-cellular localization was checked; one protein was periplasmic, one was an outer membrane protein, two had unknown localization prediction, three were cytoplasmic membrane proteins, and 31 were cytoplasmic proteins (Appendix A).

The cytoplasmic proteins (*n* = 31) were checked for conservation in *H. pylori* strains available at NCBI, using NCBI BLASTp, where three proteins showed <200 alignment scores to other *H. pylori* strains and were, therefore, excluded. The remaining proteins (*n* = 28) were checked for non-homology to the proteome of common gut flora organisms, where 11 proteins had >200 alignment scores and were excluded (Appendix A). The remaining seventeen targets included the targets retrieved from the other two approaches, besides the hypothetical protein HP0405 (Table 1; Figure 3), which had an alignment score >200 to common pathogens.

Our retrieved targets had 42 different potential binding ligands at the drugbank, after excluding repetitions and antibiotics in some targets. All ligands were classified as experimental small molecules, except for pyrophosphoric acid, citric acid, and flavin mononucleotide, which were approved small molecules. Some ligands can interact with multiple targets. 2′-monophosphoadenosine-5′-diphosphate can act as a binding ligand for AroE, LacA/Rpi, and TrbB/Vir11_2, while 2-methylthio-n6-isopentenyl-adenosine-5′-monophosphate can act on both RpsF and RpsJ. The retrieved ligands, their classes, current uses, and the expect value of the drugbank blast of the target proteins are summarized in Appendix A.

### 2.4. The In Vitro Anti-Helicobacter Activity of Selected Ligands 

Three ligands, citric acid, dipicolinic acid, and pyrophosphoric acid, were tested for their anti-helicobacter activity. Minimum inhibitory concentration (MIC) determination, by agar dilution and broth micro-dilution methods, was used to test the drug ligands against the *H. pylori* ATCC 43504 strain and a clinical *H. pylori* isolate. The three tested ligands were effective as anti-helicobacter agents. The MIC values for citric and dipicolinic acid, by both tested methods, were 1.5 mg/mL against the *H. pylori* ATCC 43504 strain and the *H. pylori* clinical isolate. The pyrophosphoric acid MIC was 3 and 2 mg/mL by agar dilution and broth micro-dilution methods, respectively, against the tested *H. pylori* isolates (Figure 4). 

## 3. Discussion

Treatment of *H. pylori* infections is becoming more difficult with the reported increase in the rate of emergence of resistance to most of the available antibiotics and the scarce development of newer effective antimicrobials. Repurposing of non-antibiotic drugs, as antimicrobials, has gained renewed attention, saving time and cost required to design, synthesize, and test novel therapeutic compounds [11]. Three different in silico approaches were used to search for promising drug targets in *H. pylori*, all of which used almost the same tools, with the same parameters, but in a different order. The selected criteria aimed to ensure efficient druggable targets with minimum host side effects. 

To avoid host toxicity, drug targets with similar counterparts in human should be excluded [14]. In all our studied approaches, non-homology to human proteome was tested with strict criteria: an expected threshold of 0.005, although the default expected threshold was set to 10 at NCBI BLASTp, and a deep scoring matrix “BLOSUM62” targeted alignments with even 20–30% identity [34]. In contrast to studies that accept targets with <100 scores [35,36,37], we excluded any targets that showed even minor significant similarity (with scores <40). All the selected drug targets were essential proteins, where essential genes/proteins are indispensable for the life of organisms, and therefore, can act as potential drug targets, especially if they are conserved [38]. Proteins with >95% identity to *H. pylori* essential proteins were selected based on a 0.00001 expected threshold and “BLOSUM62” matrix. Furthermore, the selected essential proteins were blasted against *H. pylori* strains with the highest possible alignment scores; this confirmed conservation in *H. pylori*. Also, being conserved across common pathogenic species allows broad spectrum targeting. Our targets showed significant similarity with the proteome of 228 common pathogenic organisms retrieved from a similar in silico study [29].

All targets that lacked significant similarity to the common gut flora organisms (AtpF, AtpH, FolB, NuoF, HP0276, HP0439, and HP1247) were non-druggable, thus hindering the repurposing of available agents for their inhibition. However, these proteins represent promising targets for the design of new anti-helicobacter agents. FolB, involved in folate biosynthesis, was druggable in a similar in silico study as a potential drug target for *Klebsiella pneumoniae* [39]. In our study, the cut-off value of alignment similarity scores was reset to not exceed 200, in order to minimize the disruption of the host’s gut flora as much as possible, and the subsequent adverse effects related to their possible inhibition [29]. Determination of the sub-cellular localization of candidate proteins is an essential tool for identifying effective drug targets and vaccine candidates in bioinformatics studies. Cytoplasmic proteins are favorable drug targets [40]. Targets with only antibiotic ligands were also excluded, to avoid resistance to the currently in use antibiotics and to ensure novelty of the ligands. 

In our study, 6, 15, and 17 targets were retrieved from chokepoint proteins, metabolic pathway proteins, and essential proteins analysis approaches, respectively. All essential proteins approach targets were also retrieved by the other two approaches, as they had to pass through DEG at a certain point of the analysis. Therefore, the essential proteins analysis approach is the best approach in terms of time, effort, and ease of analysis; it was previously used in the identification of potential drug targets in *Burkholderia pseudomallei* [16], and to identify potential membrane proteins as vaccine targets in *H. pylori* [23]. 

Similar to our study, KEGG and DEG databases were also used to identify drug targets in *H. pylori* strain HPAG1 [26]. CheY and Ddl were common drug targets; however, their other detected targets did not make it to our final list. This is due to having a lower essential percentage identity cut-off set value (˂95%, e.g., KdsA, GlmU, and TrpA), not being cytoplasmic (e.g., RfaC, and MurF), having a high similarity score with common gut flora (>200, e.g., LpxD, and LpxC) and being non-druggable (e.g., WaaA, and GmhB). Neelapu and co-workers used a different set of computational resources to identify drug targets in *H. pylori* [24]; most of their protein targets were excluded from our analysis for not achieving the cut-off percentage identity at DEG (e.g., RpmG, MoaD, ThiM, and ThiE), not being cytoplasmic (e.g., DppC, and HP0164), or not being druggable (e.g., Hup). 

Five protein targets, AroQ, DapB, Ddl, IspA, and RibC were commonly retrieved from our three approaches. Both AroQ and AroE (retrieved from metabolic pathway proteins and essential proteins analysis approaches) are involved in the chorismate synthesis, which is the precursor of folic acid and aromatic amino acids (phenylalanine, tyrosine, and tryptophan) that take part in protein and nucleic acid biosynthesis. Inhibition of folate biosynthesis is known to exert an anti-helicobacter activity [41,42]. DapB is involved in lysine biosynthesis via diaminopimelate, which are compounds used by bacteria to link peptidoglycan covalently to their cell wall. Ddl, with the aid of alanine racemase (Alr), builds up D-alanyl-D-alanine for peptidoglycan biosynthesis. IspA catalyses two sequential reactions in the isoprenoid or terpenoids biosynthesis, which are required in cell wall synthesis. Targeting cell wall biosynthesis by both amoxicillin and cephalosporins is already being used in the treatment of *H. pylori* infections [43]; however, these antibiotics have different enzymatic drug targets. RibC is involved in riboflavin biosynthesis, which is the precursor of the coenzymes required in redox reactions. Some antibiotics (e.g., ampicillin, gentamicin, and norfloxacin) can alter the redox homeostasis of bacterial cells, to pose an oxidative stress that contributes, ultimately, to cell death [44]. 

Targets retrieved from the metabolic pathway analysis were similar to those identified from chokepoint protein analysis, except for CmoA, which was not identified by the metabolic pathway analysis. When searching for a CmoA sequence at the KEGG database, it was not present in KEGG pathways, but was present in KEGG brite. The brite database is an ontology database that has been introduced to expand the coverage of genes for KEGG mapping [32]. CmoA converts S-adenosyl-L-methionine to carboxy-S-adenosyl-L-methionine, which is involved in the maturation of bacterial tRNAs [45]. Aminoglycosides are known to interfere with the tRNA maturation process [46], where some members of this class (gentamicin and netilmicin) have promising anti-helicobacter activity [47]. 

In addition, ten targets, AroE, CheY, PanD, TrbB/VirB11_2, DnaN, RplE, RpsF, RpsJ, LacA/Rpi, and LpxA, were retrieved from both metabolic pathway proteins analysis, and essential proteins analysis approaches. Chemotaxis protein, CheY, is involved in the signal transmission from chemoreceptors to the flagellar motors. Chemotaxis and motility are essential for both the survival and colonization of *H. pylori*, where impaired chemotactic responses to gastric mucin makes the pathogen move in a linear rather than swarming and tumbling motion, which can end up in the failure to evade gastric acidity, colonization, and allow pathogen removal by gastric flow [48]. PanD catalyzes the conversion of L-aspartate into β-alanine, which is further required for pantothenate biosynthesis, the precursor of coenzyme A. Disruption of the genes/enzymes in CoA biosynthesis can lead to lethal phenotypes [49]; also, several natural and synthetic pantothenic acid analogues possess antibacterial activity [50]. TrbB/VirB11_2 is essential for gastric colonization and represents a component of the type IV secretion system, which is required for the oncogenic CagA transport into the targeted epithelial cells [51,52]. Pathogenesis, severity of infection, malignancies, and survival in acid moiety are significantly related to the presence of a cag pathogenicity island (cag PAI) and a type IV secretion system, compared to cag PAI-negative strains [53,54,55]. Compounds with anti-virulence activity are considered unusual promising agents that help the immune system to overcome the infectious agents without development of resistance [56].

The DNA polymerase III β subunit (DnaN) and the ribosomal subunits (RplE, RpsF, and RpsJ) are involved in genetic information processing. DnaN functions as a sliding clamp during DNA replication, and is essential for cell viability, therefore representing a promising target for antibacterial drugs. The binding sites and sequences of bacterial and eukaryotes sliding clamps are different, allowing for specific targeting [57]. Our criteria allowed only for the selection of conserved targets among the tested *H. pylori* strains with an alignment score >200, and exclusion of any targets with even slight similarity to the human proteome. The non-steroidal anti-inflammatory drug, diflunisal, exerts an anti-helicobacter activity by inhibiting the β-clamp function [58]. Also, targeting DnaN was efficient in tuberculosis therapy [59]. Inhibition of ribosomal subunits interferes with protein synthesis, which is already targeted by many antibiotics, including clindamycin, puromycin, and tetracycline [60]. LacA/Rpi is involved in lipid biosynthesis, while LpxA, along with other proteins, is involved in lipopolysaccharide biosynthesis. Inhibition of lipid biosynthesis is a drug target in many bacterial species [61]. HP0405 was only identified by using the essential protein analysis approach; it is an NifS-like hypothetical protein of an unknown function [62]. 

Many binding ligands, for these targets, were identifiable in drugbank. The drugbank blast results for our protein targets showed a low expected threshold value, indicating very low chance of error hits and potential specific binding to the ligands. Some of our ligands are small organic acids with documented safety and antibacterial activity, such as (S)-3-phenyllactic acid [63], citric acid [64], malonic acid [65], dipicolinic acid [66], and D-tartaric acid [67,68]. Organic acids are used as food additives and preservatives [69,70], and have also shown anti-helicobacter activity as components in probiotics and natural products [71,72,73]. 

Nicotinamide adenine dinucleotide phosphate, an AroE binding ligand, and guanosine-5′-monophosphate, a CheY binding ligand, possess β-lactamase inhibition activity [74,75]. Both quinic and shikimic organic acid exert an inhibitory effect on *Staphylococcus aureus* [76]; we detected some of their derivatives (3-dehydroquinic acid; 2,3-anhydro-quinic acid; 3-hydroxyimino quinic acid; 2-anhydro-3-fluoro-quinic acid; 3-dehydroshikimate) as AroQ binding ligands. Most of the AroE, AroQ, CheY, DapB, Ddl, LpxA, and PanD detected binding ligands were previously reported, in a similar in silico study, to bind to the same targets in *Campylobacter jejuni* [77]. 

Searching the protein data bank revealed that seven out of the 17 protein targets (AroE, AroQ, CheY, Ddl, DnaN, LpxA, and PanD) have their crystalline 3D structure available, facilitating performing a further docking study to confirm the inhibitory role of their ligands.

We further confirmed our in silico study by testing three drug ligands, citric, dipicolinic, and pyrophosphoric acid, for their anti-helicobacter activity. The drugs were chosen based on their possible activity against three of the common protein targets, retrieved by all three approaches (RibC, DapB, and IspA). The MIC values for citric, dipicolinic, and pyrophosphoric acid ranged from 1.5–2.5 mg/mL. The MIC detected by the agar dilution and broth micro-dilution methods were comparable. The anti-helicobacter activity of citric acid was previously reported [78]. However, this is the first report of the anti-helicobacter activity of dipicolinic, and pyrophosphoric acid.

## 4. Materials and Methods 

### 4.1. Analysis of Chokepoint Proteins and Metabolic Pathway Proteins 

The initial protein pools used for these two approaches were retrieved from either the BIOCYC chokepoint reactions database [79], or KEGG pathways database [31,32,33]. This was followed by different subtractive proteomics steps to reach efficient druggable targets. A schematic flow of these two approaches is illustrated in Figure 5.

#### 4.1.1. Retrieval of Chokepoint Proteins of *H. pylori*

Chokepoint reactions of *H. pylori* ATCC 43504 were retrieved from the BIOCYC collection database [79], available at http://biocyc.org/chokepoint-form.shtml, with the following criteria: exclusion of reactions found in humans, exclusion of reactions catalyzed by more than one enzyme, and then inclusion of all reactions. Resulted reactions, from both consuming and producing sides, were checked to remove spontaneous reactions, reactions by enzymes that have not been identified by this database, and repeated reactions on both sides. 

#### 4.1.2. Retrieval of Proteins Involved in *H. pylori* Metabolic Pathways 

All the metabolic pathways of *H. pylori* strain 26695 (ATCC 700392) were retrieved from the KEGG database [31,32,33], available at https://www.genome.jp/kegg/pathway.html. Repeated reactions from the different pathways were removed, along with spontaneous reactions or those with no identifiable enzymes. The search tool BLAST FASTA (prot query vs. prot db), available at https://www.genome.jp/tools/fasta/, was used whenever needed to find a particular protein at KEGG database.

#### 4.1.3. Non-Homology Analysis to Human Proteins

Non-homology analysis of the retrieved targets was performed using the NCBI BLASTp tool (protein-protein blast) [80], available at https://blast.ncbi.nlm.nih.gov, with the non-redundant protein sequences (nr) database against *Homo sapiens* (taxid:9606), and with algorithm general parameters as an expect threshold of 0.005 and scoring parameters as matrix BLOSUM62. Any protein with significant alignments (even if <40) was excluded. 

#### 4.1.4. Essentiality Analysis

Protein targets that were not homologous to human proteome were searched for essentiality in *H. pylori* using DEG [38,81,82], version 15.2, available at http://www.essentialgene.org, with parameters set as an expected threshold of 0.00001 and matrix BLOSUM62. Any result with a percentage identity >95 was accepted. 

#### 4.1.5. Prediction of Sub-Cellular Localization

The sub-cellular localization of the retrieved essential proteins was predicted to select intracellular cytoplasmic proteins using the cellular localization prediction tool PSORTb version 3.0.2 [83], available at http://www.psort.org/psortb/. Final prediction as “cytoplasmic” with a cut-off score of >8.5 was accepted. Proteins with localizations other than cytoplasmic were excluded. 

#### 4.1.6. Conservation in *H. pylori*


The conservation of cytoplasmic targets was tested in all available *H. pylori* strains at the NCBI database (Appendix A) using the NCBI BLASTp tool, with an expected threshold of 0.0001. Only targets that were conserved with >200 alignment scores, in all tested *H. pylori* strains, were used in subsequent steps.

#### 4.1.7. Non-Homology to Proteome of Common Gut Flora Organisms

Protein targets were compared to the proteomes of common organisms known to naturally inhabit the gut of healthy individuals, according to Raman and colleagues [29], using the NCBI BLASTp tool with an expected threshold of 0.0001. 

#### 4.1.8. Druggability Analysis

Targets were searched for at Drugbank collection version 5.1.6 [30], available at http://www.drugbank.ca/, to find non-antibiotic ligands using the default BLAST parameters, with an expected threshold of 0.00001 and with the drug type filter set to include all drug types. 

#### 4.1.9. Checking for Broad-Spectrum Targeting 

Druggable targets were compared to the proteomes of common pathogenic organisms [29] using NCBI the BLASTp tool with an expect threshold of 0.0001 and matrix BLOSUM62. 

### 4.2. Essential Proteins Analysis 

A similar approach to the aforementioned approaches, but with a different flow pattern, was tested (Figure 6). All search tools, parameters, and thresholds were like those described in the previous two approaches. 

### 4.3. Determination of MIC 

The bacterial strains used in the study included *H. pylori* ATCC 43504, and a clinical *H. pylori* isolate obtained from the culture collection of the department of clinical pathology, Faculty of Medicine (Kasr El-Aini), Cairo university, Cairo, Egypt. Isolates were subcultured on Muller-Hinton agar (MAST, Bootle, UK), supplemented with 5% sheep blood and DENT supplement (Oxoid, Basingstoke, UK), and incubated at 37 °C for 72 h under microaerophilic conditions (5% O_2_, 10% CO_2_, and 85% N_2_ at 95% humidity) using CamyGen paper sachets (Oxoid, Basingstoke, UK) [84]. Colonies were suspended in saline to reach an optical density equivalent to McFarland 2.0 turbidity standard (approximately 1 × 10^7^–1 × 10^8^ CFU/mL) [85]. 

Citric acid (Loba Chemie, Mumbai, India), dipicolinic acid (Alfa Aesar, Kandel, Germany), and pyrophosphoric acid (Sigma Aldrich, Steinheim, Germany) solutions in distilled water were freshly prepared and sterilized by membrane filtration using a 0.22 µm pore-size syringe filter (StarTech, Northampton, UK). The final tested concentrations ranged from 0.19–6 mg/mL. The MIC values were determined using the agar dilution and broth micro-dilution methods.

The agar dilution method was performed according to the Clinical and Laboratory institute CLSI guidelines. Briefly, 2 µL inoculums (equivalent to 1 × 10^4^ CFU/spot) were delivered to the surface of plates of Muller-Hinton agar supplemented with 5% sheep blood and containing the specified dilution of the ligand. Plates were incubated at 37 °C for 72 h, under microaerophilic conditions. The experiment was done in triplicates. Inoculum was delivered to the surface of Muller-Hinton agar plates supplemented with 5% sheep blood to act as growth control. The MIC value was the lowest concentration of the compound, completely inhibiting visible bacterial growth [85].

The broth micro-dilution method was performed, according to Piccolomini and colleagues [84], with minor modifications. Briefly, ligand solutions were diluted in Brucella broth (Conda, Madrid, Spain) supplemented with 10% fetal bovine serum (Sigma-Aldrich, Steinheim, Germany) to get the specified concentrations. Each well contained 100 µL of ligand-containing broth at the specified concentration. The adjusted inoculum (10 µL) was added, so that each well contained 5 × 10^5^ CFU/mL, and the microtiter plates were incubated at 37 °C for 72 h under microaerophilic conditions. The MIC was the lowest concentration that completely inhibited the visible growth of the tested organism.

## 5. Conclusions

Using subtractive proteomics approaches proved very efficient in saving time, money, and effort expended in the detection of novel drug targets and their potential inhibitors. This approach enabled us to identify 17 potential druggable targets in *H. pylori*, and the possible repurposing of available agents to inhibit these targets. This provides new hope for saving lives of those at high risk of infection with the carcinogenic *H. pylori* pathogen. This is a preliminary study; further testing is still required to confirm the potential use of these ligands in the treatment of *H. pylori* infections and the target specificity, as well as the safety and possible side effects.

## Figures and Tables

**Figure 1 pathogens-09-00747-f001:**
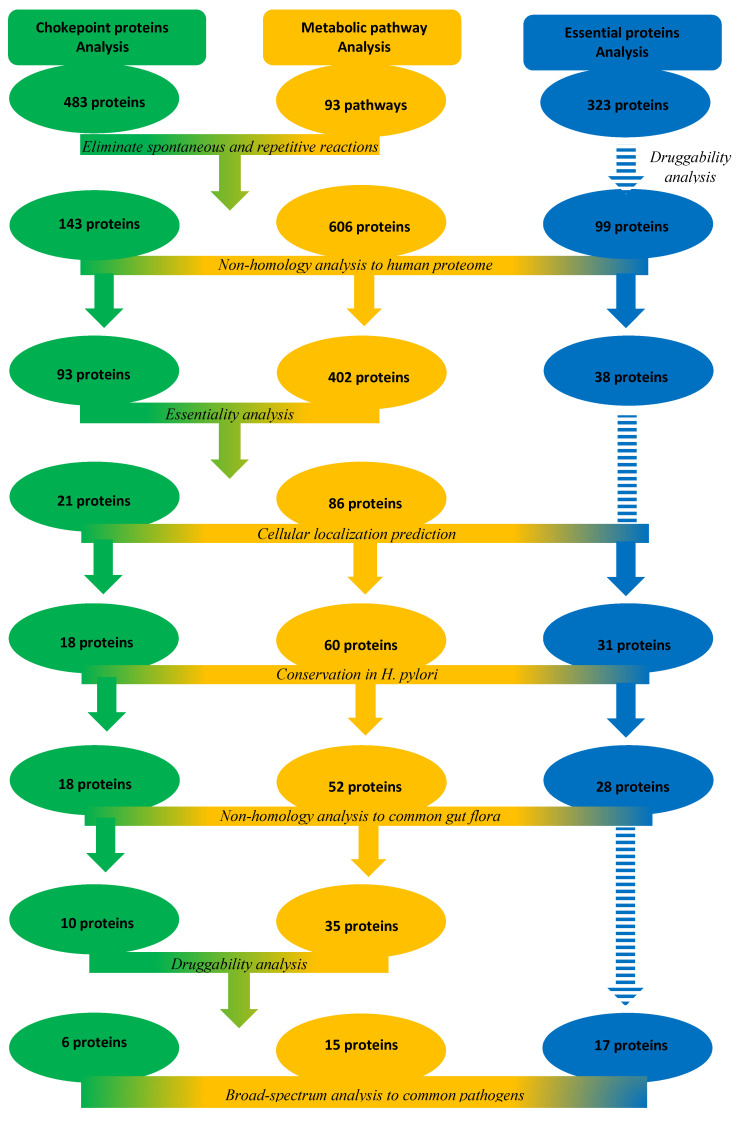
The three subtractive proteomics approaches used to identify potential drug targets in *H. pylori* proteome and their outcomes.

**Figure 2 pathogens-09-00747-f002:**
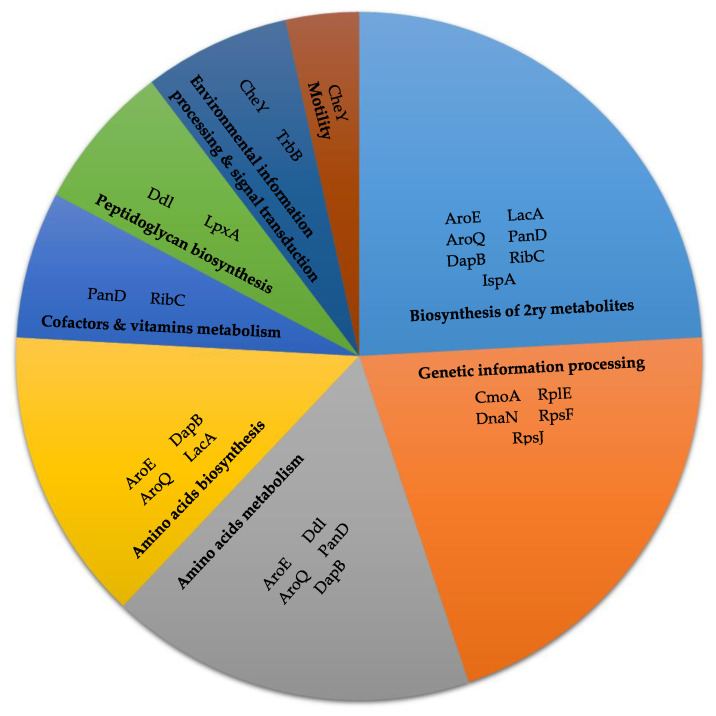
Pathways of targets retrieved from both chokepoint proteins analysis and metabolic pathway proteins analysis.

**Figure 3 pathogens-09-00747-f003:**
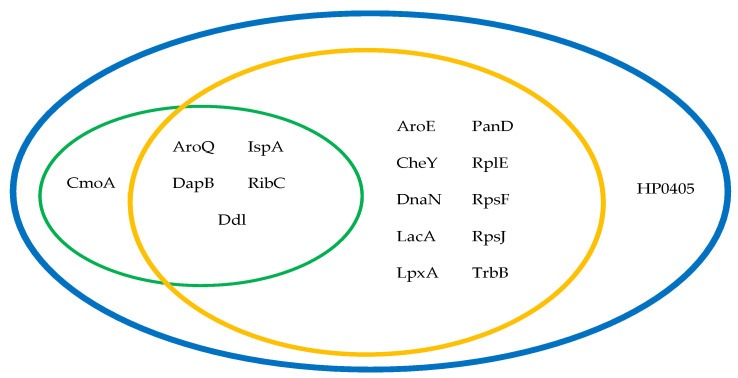
Protein targets retrieved from chokepoint proteins (green), metabolic pathway (orange), and essential proteins analysis (blue) approaches.

**Figure 4 pathogens-09-00747-f004:**
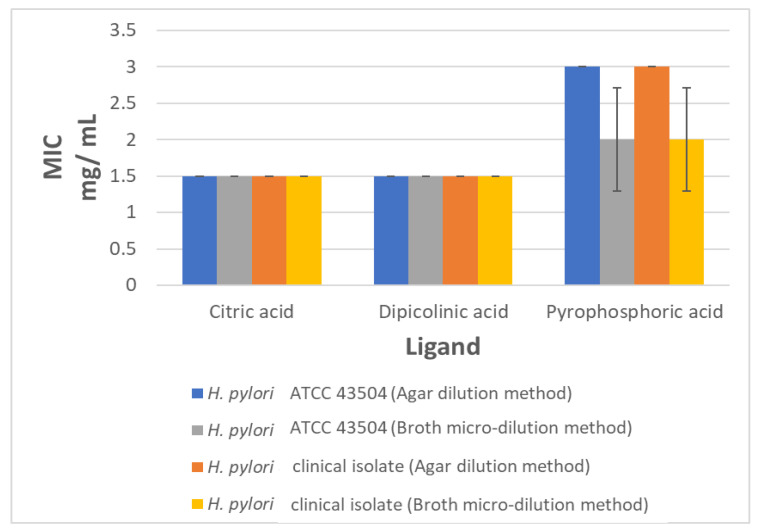
Minimum inhibitory concentration (MIC) of the tested ligands against tested *H.pylori* strains using agar dilution and broth micro-dilution methods. MIC values represent the mean of three experimental repliactes.

**Figure 5 pathogens-09-00747-f005:**
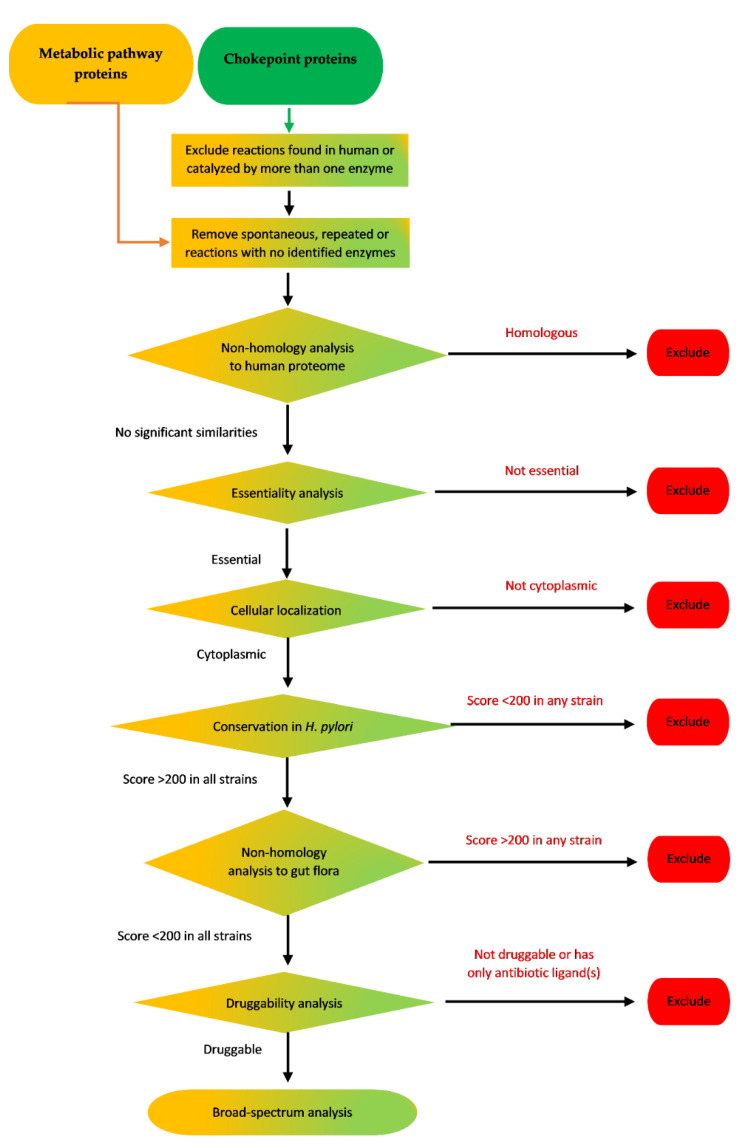
Schematic flow chart of chokepoint proteins analysis and metabolic pathway proteins analysis of *H. pylori* proteome.

**Figure 6 pathogens-09-00747-f006:**
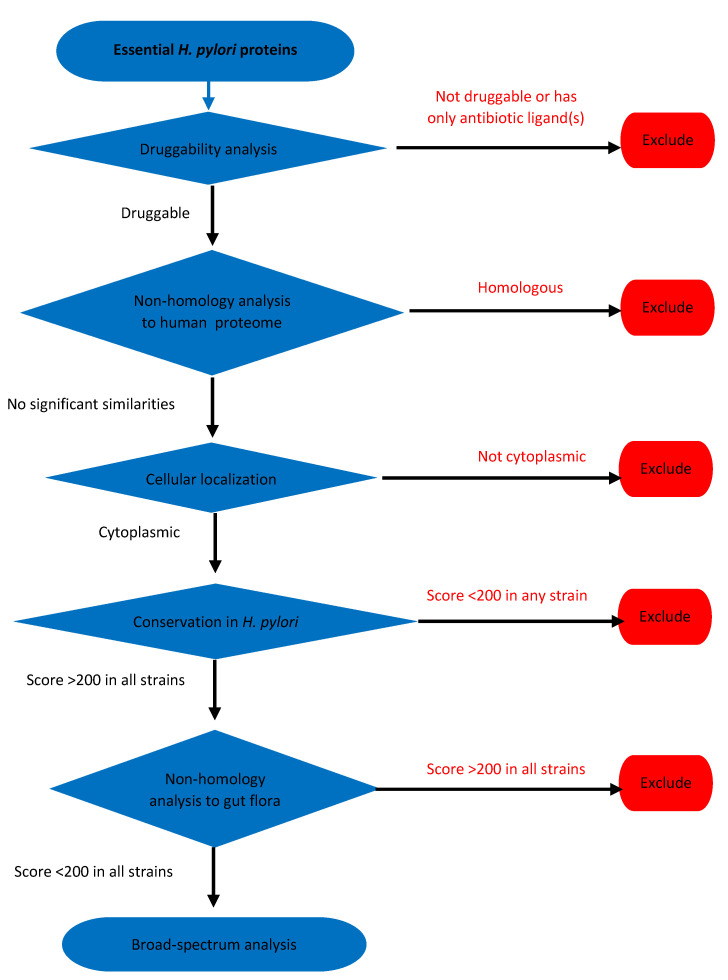
Schematic flow chart of essential proteins analysis of *H. pylori* proteome.

**Table 1 pathogens-09-00747-t001:** Characteristics of *H. pylori* protein targets retrieved from chokepoint proteins, metabolic pathways and essential proteins subtractive approaches.

Target	Protein Name	App	Pathway	Similarity to Common Pathogen	Possible Ligands	Ligand Drugbank Accession Number
AroE	Shikimate dehydrogenase	PW EP	Phenylalanine, tyrosine, and tryptophan biosynthesis	Mostly 80–200	1,4-Dithiothreitol	DB04447
					Nicotinamide adenine dinucleotide phosphate	DB03461
					2′-Monophosphoadenosine-5′-diphosphate	DB02363
AroQ	3-dehydroquinate dehydratase	CP PW EP	Phenylalanine, tyrosine, and tryptophan biosynthesis	Mostly 80–200	3-Dehydroquinic Acid	DB03868
					N-(1,4-dihydro-5H-tetrazol-5-ylidene)-9-oxo-9H-xanthene-2-sulfonamide	DB04698
					2,3-Anhydro-quinic acid	DB02801
					3-Hydroxyimino quinic acid	DB03739
					2-Anhydro-3-fluoro-quinic acid	DB02786
					3-Dehydroshikimate	DB04347
					1,3,4-Trihydroxy-5-(3-phenoxypropyl)-cyclohexane-1-carboxylic acid	DB04656
					(1s,4s,5s)-1,4,5-Trihydroxy-3-[3-(phenylthio)phenyl]cyclohex-2-ene-1-carboxylic acid	DB08485
CheY	chemotaxis protein	PW EP	Two-component system and bacterial chemotaxis	>200	S-Methyl Phosphocysteine	DB02461
					(S)-Aspartimide	DB03487
					Aspartate Beryllium Trifluoride	DB04156
					Adenosine-5′-Rp-Alpha-Thio-Triphosphate	DB02355
					alpha,beta-Methyleneadenosine 5′-triphosphate	DB02596
					2-Hydroxyestradiol	DB07706
					Guanosine-5′-Monophosphate	DB01972
					Phosphoaspartate	DB01857
CmoA	Carboxy-S-adenosyl-L-methionine synthase	CP EP	5-(methoxycarbonylmethoxy)uridine biosynthesis	>200	S-Adenosyl-L-Homoselenocysteine	DB03423
DapB	4-hydroxy-tetrahydrodipicolinate reductase	CP PW EP	Lysine biosynthesis	>200	3-Acetylpyridine Adenine Dinucleotide	DB03363
					Dipicolinic acid	DB04267
Ddl	D-alanine-D-alanine ligase	CP PW EP	D-alanine metabolism and Peptidoglycan biosynthesis	>200	3-Chloro-2,2-dimethyl-n-[4-(trifluoromethyl)phenyl]propanamide	DB07805
DnaN	DNA polymerase III subunit β	PW EP	DNA replication, Mismatch repair, and Homologous recombination	>200	[(5R)-5-(2,3-dibromo-5-ethoxy-4-hydroxybenzyl)-4-oxo-2-thioxo-1,3-thiazolidin-3-yl]acetic acid	DB06998
HP0405	Hypothetical protein 0405	EP	NA	>200	Selenocysteine	DB02345
					S-Mercaptocysteine	DB02761
					S-Selanyl Cysteine	DB03049
					L-2-amino-3-butynoic acid	DB04217
					3′-O-N-Octanoyl-a-D-Glucopyranosyl-B-D-Fructofuranoside	DB02346
IspA	Geranyl diphosphate synthase	CP PW EP	Terpenoid backbone biosynthesis	>200	Pyrophosphoric acid	DB04160
					Isopentyl Pyrophosphate	DB02508
					Dimethylallyl S-Thiolodiphosphate	DB02270
LacA/ Rpi	Ribose-5-phosphate isomerase B	PW EP	Pentose phosphate pathway, Fructose and mannose metabolism	80–200	2′-Monophosphoadenosine-5′-Diphosphate	DB02363
LpxA	Acyl-[acyl-carrier-protein]—UDP-N-acetylglucosamine O-acyltransferase	PW EP	Lipopolysaccharide biosynthesis	>200	D-tartaric acid	DB01694
					2-Hydroxymethyl-6-octylsulfanyl-tetrahydro-pyran-3,4,5-triol	DB08558
					4-Chloro-N-(3-methoxypropyl)-N-[(3S)-1-(2-phenylethyl)piperidin-3-yl]benzamide	DB08344
PanD	Aspartate 1-decarboxylase proenzyme	PW EP	β-alanine metabolism	80–200	Malonic acid	DB02175
					S-oxy-L-cysteine	DB03382
RibC	Riboflavin synthase	CP PW EP	Riboflavin metabolism	>200	Citric acid	DB04272
					Flavin mononucleotide	DB03247
					Lumichrome	DB04345
RplE	50S ribosomal subunit L5	PW EP	Ribosome	>200	(S)-3-phenyllactic acid	DB02494
RpsF	30S ribosomal subunit S6	PW EP	Ribosome	80–200	2-Methylthio-n6-isopentenyl-adenosine-5′-monophosphate	DB08185
RpsJ	30S ribosomal subunit S10	PW EP	Ribosome	80–200	2-Methylthio-n6-isopentenyl-adenosine-5′-monophosphate	DB08185
TrbB/VirB11_2	Type IV secretion system ATPase	PW EP	Bacterial secretion system Epithelial cell signaling	Mostly 80–200	2′-Monophosphoadenosine-5′-Diphosphate	DB02363

App = Approach; CP = Chokepoint proteins approach; PW = Metabolic pathway proteins approach; EP = Essential proteins approach.

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
