# Peer review of "Identification of Potential Drug Targets in Helicobacter pylori Using In Silico Subtractive Proteomics Approaches and Their Possible Inhibition through Drug Repurposing"

_pathogens, 2020, doi:10.3390/pathogens9090747_

Round 1
Reviewer 1 Report
In their manuscript “Identification of Potential Drug Targets in Helicobacter pylori Using In silico Subtractive Proteomics Approaches and their Possible Inhibition through Drug Repurposing” Ibrahim et al present three methodologies to determine potentially druggable Helicobacter pylori proteins. They also take it a step further and identify inhibitors to these drug targets and evaluate existing therapeutics for repurposing in Helicobacter treatment.
The presented manuscript attempts to tackle an urgent clinical need and is therefore of high academic and societal relevance. Three different subtractive proteomic approaches yielded 17 potential protein drug targets. The different methods are compared, and the DEG approach clearly outperformed the other two, chokepoint reactions in BIOCYC and KEGG. The authors were also able to identify 42 binding ligands that could serve as inhibitors and are partially already used in the clinics, which would make it easy to repurpose them for H. pylori treatment.
The manuscript is well written and the approaches applied sound as well as the thresholds applied stringent. I recommend accepting the manuscript for publication in Pathogens after addressing the following minor remarks:
Line 25: “The use of the initial protein pools from DEG was superior, to the other two in terms of retrieval of nearly all drug targets identified by them.” In figure 3 you actually show that DEG does not only cover nearly all, but actually all of the combined drug targets. Adjust accordingly.
Line 39-40: This claim is a) not correct (gastric cancer was the second most common cause of death amongst cancer types) and b) not timely enough since it is from 2008. Also cite the original paper next time.
Line 55: Add drugs or agents after anti-psychotic
Line 79a: protein pools
Line 79b: metabolic pathway proteins
Figure 1: Clean up the figure in terms of the positioning; the ovals are smaller probably to indicate reduction in number, but they should be place always in equal distance to the respective
Line 84: Why was this particular strain chosen? Is it of particular clinical relevance?
Line 99: Which common gut flora organisms are included? Please provide this information in an easy-to-find way.
Line 102: Please cite drugbank also right here and not only later.
Table 1: Try to condense this table. Make the two leftmost columns narrower, reduce the spaces between the columns and/or the spaces between text and the border of the column. Also another table, potentially in the supplementary, would be nice with an overview of the potential ligands, their molecule class, current uses within and outside the clinics and whether they are already in use against Helicobacter.
Line 110: Also cite KEGG here.
Line 119: tohaving -> to having
Figure 2: LpxA and DapB are truncated -> correct this; Also the Biosynthesis of secondary metabolites writing could be put horizontally for easier reading
Figure 3: Indicate the color coding in text form the figure caption, rather than with the colors below the caption. Also use the same line width.
Line 151: Remove capitalization for flavin
Line 154: Again remove capitalization for methylthio
Line 160: Space missing between [11]. And Three
Line 188: Rephrase “the already in use” to current or similar
Line 190: Please use a number here for ease of reading and don't write the number out
Line 199: Rephrase “failed to pass to our final list”, eg with did not make it into our final list
Line 211-215: Try to break down this very long sentence into several sentences for ease of reading.
Reviewer 2 Report
In this study, Ibrahim and collaborators sought to identify novel drug targets to control H. pylori infection. To this end, the authors employ three different subtractive approaches, which retrieve a total of 17 proteins as potential druggable targets.
The identification of novel treatment strategies against H. pylori infection is extremely important, considering the worrying increase in antibiotic-resistant strains. Using an in silico approach the authors come up with a very short list of bacterial proteins that can be targeted by known drugs. This is very interesting, however also preliminary overall.
Some of the proteins identified, as AroE, are certainly good targets. However, other seem not to be too specific, as DnaN, since DNA polymerases tend to be extremely well conserved, while others seem not to be too relevant, such as TrbB, as not all H. pylori strains possess the cag pathogenicity island where the type IV secretion system is encoded. How was this taken into account?
The relevance of the reported results needs to be confirmed experimentally. The confirmation of a couple of the identified proteins as druggable targets would be important to validate their approach.
Round 2
Reviewer 2 Report
The authors have addressed some points in the revised version of the manuscript. However, the last point is still unclear. The authors have performed an experiment to determine the effect of 3 drugs on the growth of H. pylori. How specific are those drugs to the corresponding identified targets? This has not been addressed. Also, the results reported on inhibition of bacterial growth are not shown in a corresponding figure. A figure including controls to show specificity of the substances used should be included in the manuscript.
Author Response
Dear Reviewer,
We would like to thank you for taking the time and effort necessary to review the manuscript. We sincerely appreciate all your valuable comments and suggestions, which helped us to improve the quality of the manuscript.
Comment to authors:
The authors have addressed some points in the revised version of the manuscript. However, the last point is still unclear. The authors have performed an experiment to determine the effect of 3 drugs on the growth of H. pylori. How specific are those drugs to the corresponding identified targets? This has not been addressed. Also, the results reported on inhibition of bacterial growth are not shown in a corresponding figure. A figure including controls to show specificity of the substances used should be included in the manuscript.
-Figure 4 showing the results of bacterial inhibition of growth was added in line 184, and the numbering of the subsequent figures was updated.
-Regarding the specificity of targets: the drugbank blast results for our protein targets showed a low expect threshold value, indicating a very low chance of error hits and potential specific binding to the ligands, this was explained in discussion section lines 297-299, and the expect values were added to supplementary table 5, and line 174. The E value of the drugbank blast for RibC (citric acid as a binding ligand), IspA (pyrophosphoric acid as a binding ligand), DapB ( dipicolinic acid as a binding ligand) were 1.76801E-18, 1.41774E-38, and 6.14853E-31, respectively. This indicates a very low chance of error hits and hence specificity of targets. Our study is a preliminary one. Testing the specificity of targets is beyond the scope of the current manuscript that describes preliminary data that needs intensive invitro and in vivo validation. This has been clarified in the conclusion section lines 417-420.
Round 3
Reviewer 2 Report
The authors have now included a graph were they show the MIC values of the different substances tested. However, and as indicated in the previous revision, authors should show the growth curves of the bacteria from which the MICs were calculated and not merely the MIC values.
Author Response
Dear reviewer
Thank you so much for carefully reviewing our manuscript and your valuable comments.
Reviewer comment:
The authors have now included a graph where they show the MIC values of the different substances tested. However, and as indicated in the previous revision, authors should show the growth curves of the bacteria from which the MICs were calculated and not merely the MIC values.
In our study, the minimum inhibitory concentration (MIC) was determined by two established methods: the agar dilution and broth micro-dilution methods, as stated in the manuscript lines 282-295. The Agar dilution method is the CLSI recommended method for the determination of MIC against H. pylori. The procedure and recommendation from CLSI are cited in the text. Accordingly, no growth curve was used, and the MIC value was determined as the lowest concentration of the compound, which completely inhibiting visible bacterial growth. Similarly, the broth micro-dilution method was also employed for MIC determination. Both methods had comparable results (lines 64-68). The broth microdilution method reference is cited in line 289. No growth curve was employed for MIC determination as the MIC value was the lowest concentration that completely inhibited the visible growth of the tested organism. CLSI recommendations for MIC determination by broth methods involve reading the MIC results by an unaided eye. Consequently, no growth curves were employed for MIC calculation to be added to the manuscript.
Regards,
Round 4
Reviewer 2 Report
The authors have addressed all points.